# On Power Laws in Deep Ensembles

**Ekaterina Lobacheva[1], Nadezhda Chirkova[1], Maxim Kodryan[1], Dmitry Vetrov[1,2]**
[1]Samsung-HSE Laboratory, National Research University Higher School of Economics
[2]Samsung AI Center Moscow
Moscow, Russia
{elobacheva,nchirkova,mkodryan,dvetrov}@hse.ru

## Abstract

Ensembles of deep neural networks are known to achieve state-of-the-art performance in uncertainty estimation and lead to accuracy improvement. In this work, we focus on a classification problem and investigate the behavior of both non-calibrated and calibrated negative log-likelihood (CNLL) of a deep ensemble as a function of the ensemble size and the member network size. We indicate the conditions under which CNLL follows a power law w.r.t. ensemble size or member network size, and analyze the dynamics of the parameters of the discovered power laws. Our important practical finding is that one large network may perform worse than an ensemble of several medium-size networks with the same total number of parameters (we call this ensemble a memory split). Using the detected power law-like dependencies, we can predict (1) the possible gain from the ensembling of networks with given structure, (2) the optimal memory split given a memory budget, based on a relatively small number of trained networks.

## 1   Introduction

Neural networks provide state-of-the-art results in a variety of machine learning tasks, however, several neural network's aspects complicate their usage in practice, including overconfidence [18], vulnerability to adversarial attacks [30], and overfitting [29]. One of the ways to compensate these drawbacks is using deep ensembles, i.e. the ensembles of neural networks trained from different random initialization [18]. In addition to increasing the task-specific metric, e.g. accuracy, the deep ensembles are known to improve the quality of uncertainty estimation, compared to a single network. There is yet no consensus on how to measure the quality of uncertainty estimation. Ashukha et al. [2] consider a wide range of possible metrics and show that the calibrated negative log-likelihood (CNLL) is the most reliable one because it avoids the majority of pitfalls revealed in the same work.

Increasing the size $n$ of the deep ensemble, i.e. the number of networks in the ensemble, is known to improve the performance [2]. The same effect holds for increasing the size $s$ of a neural network, i.e. the number of its parameters. Recent works [3, 22] show that even in an extremely overparameterized regime, increasing $s$ leads to a higher quality. These works also mention a curious effect of non-monotonicity of the test error w.r.t. the network size, called double descent behaviour.

In figure 1, left, we may observe the saturation and stabilization of quality with the growth of both the ensemble size $n$ and the network size $s$. The goal of this work is to study the asymptotic properties of CNLL of deep ensembles as a function of $n$ and $s$. We investigate under which conditions and w.r.t. which dimensions the CNLL follows a power law for deep ensembles in practice. In addition to the horizontal and vertical cuts of the diagram shown in figure 1, left, we also study its diagonal direction, which corresponds to the increase of the total parameter count.

The power-law behaviour of deep ensembles has previously been touched in the literature. Geiger et al. [8] consider simple shallow architectures and reason about the power-law behaviour of the test

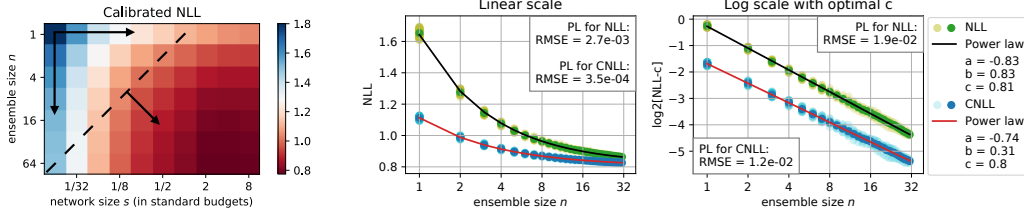

Figure 1: Non-calibrated NLL and CNLL of VGG on CIFAR-100. Left: the $(n, s)$-plane for the CNLL. Middle and right: non-calibrated $\mathrm{NLL}_n$ and $\mathrm{CNLL}_n$ can be closely approximated with a power law (VGG of the commonly used size as an example) .

error of a deep ensemble as a function of $n$ when $n \to \infty$, and of a single network as a function of $s$ when $s \to \infty$. Kaplan et al. [13], Rosenfeld et al. [26] investigate the behaviour of *single* networks of modern architectures and empirically show that their NLL and test error follow power laws w. r. t. the network size $s$. In this work, we perform a broad empirical study of power laws in deep ensembles, relying on the practical setting with properly regularized, commonly used deep neural network architectures. Our main contributions are as follows:

1. for the practically important scenario with NLL calibration, we derive the conditions under which CNLL of an ensemble follows a power law as a function of $n$ when $n \to \infty$;

2. we empirically show that, in practice, the following dependencies can be closely approximated with a power law on the *whole* considered range of their arguments: (a) CNLL of an ensemble as a function of the ensemble size $n \geqslant 1$; (b) CNLL of a single network as a function of the network size $s$; (c) CNLL of an ensemble as a function of the total parameter count;

3. based on the discovered power laws, we make several practically important conclusions regarding the use of deep ensembles in practice, e. g. using a large single network may be less beneficial than using a so-called memory split — an ensemble of several medium-size networks of the same total parameter count;

4. we show that using the discovered power laws for $n \geqslant 1$, and having a small number of trained networks, we can predict the CNLL of the large ensembles and the optimal memory split for a given memory budget.

**Definitions and notation.** In this work, we treat a power law as a family of functions $\mathrm{PL}_m = c + bm^a$, $m = 1, 2, 3, \ldots$; $a < 0$, $b \in \mathbb{R}$, $c \in \mathbb{R}$ are the parameters of the power law. Parameter $c = \lim_{m \to \infty}(c + bm^a) = \lim_{m \to \infty} \mathrm{PL}_m \stackrel{\mathrm{def}}{=} \mathrm{PL}_\infty$ reflects the asymptote of the power law. Parameter $b = c - \mathrm{PL}_1 = \mathrm{PL}_\infty - \mathrm{PL}_1$ reflects the difference between the starting point of the power law and its asymptote. Parameter $a$ reflects the speed of approaching the asymptote. In the rest of the work, $(\mathrm{C})\mathrm{NLL}_m$ denotes (C)NLL as a function of $m$.

## 2   Theoretical view

The primary goal of this work is to perform the empirical study of the conditions under which NLL and CNLL of deep ensembles follow a power law. Before diving into a discussion about our empirical findings, we first provide a theoretical motivation for anticipating power laws in deep ensembles, and discuss the applicability of this theoretical reasoning to the practically important scenario with calibration.

We begin with a theoretical analysis of the non-calibrated NLL of a deep ensemble as a function of ensemble size $n$. Assume that an ensemble consists of $n$ models that return independent identically distributed probabilities $p^*_{\mathrm{obj},i} \in [0, 1]$, $i = 1, \ldots, n$ of the correct class for a single object from the dataset $\mathcal{D}$. Hereinafter, operator $*$ denotes retrieving the prediction for the correct class. We introduce

the *model-average* NLL of an ensemble of size $n$ for the *given object*:

$$\text{NLL}_n^{\text{obj}} = -\mathbb{E} \log \left( \frac{1}{n} \sum_{i=1}^{n} p_{\text{obj},i}^* \right). \tag{1}$$

The expectation in (1) is taken over all possible models that may constitute the ensemble (e. g. random initializations). The following proposition describes the asymptotic power-law behavior of $\text{NLL}_n^{\text{obj}}$ as a function of the ensemble size.

**Proposition 1** *Consider an ensemble of $n$ models, each producing independent and identically distributed probabilities of the correct class for a given object: $p_{\text{obj},i}^* \in [\epsilon_{\text{obj}}, 1]$, $\epsilon_{\text{obj}} > 0$, $i = 1, \ldots, n$. Let $\mu_{\text{obj}} = \mathbb{E} p_{\text{obj},i}^*$ and $\sigma_{\text{obj}}^2 = \mathbb{D} p_{\text{obj},i}^*$ be, respectively, the mean and variance of the distribution of probabilities. Then the model-average NLL of the ensemble for a single object can be decomposed as follows:*

$$\text{NLL}_n^{\text{obj}} = \text{NLL}_\infty^{\text{obj}} + \frac{1}{n} \frac{\sigma_{\text{obj}}^2}{2\mu_{\text{obj}}^2} + \mathcal{O}\left(\frac{1}{n^2}\right). \tag{2}$$

*where $\text{NLL}_\infty^{\text{obj}} = -\log(\mu_{\text{obj}})$ is the "infinite" ensemble NLL for the given object.*

The proof is based on the Taylor expansions for the moments of functions of random variables, we provide it in Appendix A.1. The assumption about the lower limit of model predictions $\epsilon_{\text{obj}} > 0$ is necessary for the accurate derivation of the asymptotic in (2). We argue, however, that this condition is fulfilled in practice as real softmax outputs of neural networks are always positive and separated from zero.

The model-average NLL of an ensemble of size $n$ on the whole dataset, $\text{NLL}_n$, can be obtained via summing $\text{NLL}_n^{\text{obj}}$ over objects, which implies that $\text{NLL}_n$ also behaves as $c + bn^{-1}$, where $c, b > 0$ are constants w. r. t. $n$, as $n \to \infty$. However, for the finite-range $n$, the dependency in $\text{NLL}_n$ may be more complex.

Ashukha et al. [2] emphasize that the comparison of the NLLs of different models with suboptimal softmax temperature may lead to an arbitrary ranking of the models, so the comparison should only be performed after *calibration*, i. e. with optimally selected temperature $\tau$. The model-average CNLL of an ensemble of size $n$, measured on the whole dataset $\mathcal{D}$, is defined as follows:

$$\text{CNLL}_n = \mathbb{E} \min_{\tau > 0} \left\{ -\sum_{\text{obj} \in \mathcal{D}} \log \bar{p}_{\text{obj},n}^*(\tau) \right\}, \tag{3}$$

where the expectation is also taken over models, and $\bar{p}_{\text{obj},n}(\tau) \in [0,1]^K$ is the distribution over $K$ classes output by the ensemble of $n$ networks with softmax temperature $\tau$. Ashukha et al. [2] obtain this distribution by averaging predictions $p_{\text{obj},i} \in [0,1]^K$ of the member networks $i = 1, \ldots, n$ for a given object and applying the temperature $\tau > 0$ on top of the ensemble: $\bar{p}_{\text{obj},n}(\tau) = \text{softmax}\left\{ \left(\log(\frac{1}{n} \sum_{i=1}^{n} p_{\text{obj},i})\right) / \tau \right\}$. This is a native way of calibrating, in a sense that we plug the ensemble into a standard procedure of calibrating an arbitrary model. We refer to the described calibration procedure as applying temperature *after* averaging. In our work, we also consider another way of calibrating, namely applying temperature *before* averaging: $\bar{p}_{\text{obj},n}(\tau) = \frac{1}{n} \sum_{i=1}^{n} \text{softmax}\{\log(p_{\text{obj},i})/\tau\}$. The two calibration procedures perform similarly in practice, in most of the cases the second one performs slightly better (see Appendix C.1).

The following series of derivations helps to connect the non-calibrated and calibrated NLLs. If we fix some $\tau > 0$ and apply it *before* averaging, $\bar{p}_{\text{obj},n}(\tau)$ fits the form of the ensemble in the right-hand side of equation (1), and according to Proposition 1, we obtain that the model-average NLL of an $n$-size ensemble with fixed temperature $\tau$, $\text{NLL}_n(\tau)$, follows a power law w. r. t. $n$ as $n \to \infty$. Applying $\tau$ *after* averaging complicates the derivation, but the same result is generally still valid, see Appendix A.2. However, the parameter $b$ of the power law may become negative for certain values of $\tau$. In contrast, when we apply $\tau$ before averaging, $b$ always remains positive, see eq. (2).

Minimizing $\text{NLL}_n(\tau)$ w. r. t. $\tau$ results in a lower envelope of the (asymptotic) power laws:

$$\text{LE-NLL}_n = \min_{\tau > 0} \text{NLL}_n(\tau), \quad \text{NLL}_n(\tau) \overset{n \to \infty}{\sim} \text{PL}_n. \tag{4}$$

The lower envelope of power laws also follows an (asymptotic) power law. Consider for simplicity a finite set of temperatures $\{\tau_1, \ldots, \tau_T\}$, which is the conventional practical case. As each of $\mathrm{NLL}_n(\tau_t), t = 1, \ldots, T$ converges to its asymptote $c(\tau_t)$, there exists an optimal temperature $\tau_{t^*}$ corresponding to the lowest $c(\tau_{t^*})$. The above implies that starting from some point $n$, $\mathrm{LE\text{-}NLL}_n$ will coincide with $\mathrm{NLL}_n(\tau_{t^*})$ and hence follow its power law. We refer to Appendix A.3 for further discussion on continuous temperature.

Substituting the definition of $\mathrm{NLL}_n(\tau)$ into (4) results in:

$$\mathrm{LE\text{-}NLL}_n = \min_{\tau > 0} \mathbb{E}\left\{ -\sum_{\mathrm{obj} \in \mathcal{D}} \log \bar{p}^*_{\mathrm{obj},n}(\tau) \right\}, \tag{5}$$

from which we obtain that the only difference between $\mathrm{LE\text{-}NLL}_n$ and $\mathrm{CNLL}_n$ is the order of the minimum operation and the expectation. Although this results in another calibration procedure than the commonly used one, we show in Appendix D that the difference between the values of $\mathrm{LE\text{-}NLL}_n$ and $\mathrm{CNLL}_n$ is negligible in practice. Conceptually, applying expectation inside the minimum is also a reasonable setting: in this case, when choosing the optimal $\tau$, we use the more reliable estimate of the NLL of the $n$-size ensemble with temperature $\tau$. This setting is not generally considered in practice, since it requires training several ensembles and, as a result, is more computationally expensive. In the experiments we follow the definition of CNLL (3) to consider the most practical scenario.

To sum up, in this section we derived an asymptotic power law for LE-NLL that may be treated as another definition of CNLL, and that closely approximates the commonly used CNLL in practice.

## 3   Experimental setup

We conduct our experiments with convolutional neural networks, WideResNet [33] and VGG16 [27], on CIFAR-10 [16] and CIFAR-100 [17] datasets. We consider a wide range of network sizes $s$ by varying the width factor $w$: for VGG / WideResNet, we use convolutional layers with $[w, 2w, 4w, 8w] / [w, 2w, 4w]$ filters, and fully-connected layers with $8w / 4w$ neurons. For VGG / WideResNet, we consider $2 \leqslant w \leqslant 181 / 5 \leqslant w \leqslant 453$; $w = 64 / 160$ corresponds to a standard, commonly used, configuration with $s_{\mathrm{standard}}$ = 15.3M / 36.8M parameters. These sizes are later referred to as the standard budgets. For each network size, we tune hyperparameters (weight decay and dropout) using grid search. We train all networks for 200 epochs with SGD with an annealing learning schedule and a batch size of 128. We aim to follow the practical scenario in the experiments, so we use the definition CNLL (3), not LE-NLL (4). Following [2], we use the "test-time cross-validation" to compute the CNLL. We apply the temperature before averaging, the motivation for this is given in section 4. More details are given in Appendix B.

For each network size $s$, we train at least $\ell = \max\{N, 8s_{\mathrm{standard}}/s\}$ networks, $N = 64 / 12$ for VGG / WideReNet. For each $(n, s)$ pair, given the pool of $\ell$ trained networks of size $s$, we construct $\lfloor \frac{\ell}{n} \rfloor$ ensembles of $n$ distinct networks. The NLLs of these ensembles have some variance, so in all experiments, we average NLL over $\lfloor \frac{\ell}{n} \rfloor$ runs. We use these values to approximate NLL with a power law along the different directions of the $(n, s)$-plane. For this, we only consider points that were averaged over at least three runs.

**Approximating sequences with power laws.**   Given an arbitrary sequence $\{\hat{y}_m\}$, $m = 1, \ldots, M$, we approximate it with a power law $\mathrm{PL}_m = c + bm^a$. In the rest of the work, we use the hat notation $\hat{y}_m$ to denote the observed data, while the value without hat, $y_m$, denotes $y$ as a function of $m$. To fit the parameters $a$, $b$, $c$, we solve the following optimization task using BFGS:

$$(a, b, c) = \operatorname*{argmin}_{a,b,c} \frac{1}{M} \sum_{m=1}^{M} \left( \log_2(\hat{y}_m - c) - \log_2(bm^a) \right)^2. \tag{6}$$

We use the logarithmic scale to pay more attention to the small differences between values $\hat{y}_m$ for large $m$. For a fixed $c$, optimizing the given loss is equivalent to fitting the linear regression model with one factor $\log_2 m$ in the space $\log_2 m$ — $\log_2(y_m - c)$ (see fig. 1, right as an example).

# 4 NLL as a function of ensemble size

In this section, we would like to answer the question, whether the NLL as the function of ensemble size can be described by a power law in practice. We consider both calibrated NLL and NLL with a fixed temperature. To answer the stated question, we fit the parameters $a$, $b$, $c$ of the power law on the points $\widehat{\mathrm{NLL}}_n(\tau)$ or $\widehat{\mathrm{CNLL}}_n$, $n = 1, 2, 3, \ldots$, using the method described in section 3, and analyze the resulting parameters and the quality of the approximation.

As we show in Appendix C.2, when the temperature is applied *after* averaging, $\mathrm{NLL}_n(\tau)$ is, in some cases, an increasing function of $n$. As for CNLL, we found settings when $\mathrm{CNLL}_n$ with the temperature applied *after* averaging is not a convex function for small $n$, and as a result, cannot be described by a power law. In the rest of the work, we apply temperature *before* averaging, as in this case, both $\mathrm{NLL}_n(\tau)$ and $\mathrm{CNLL}_n$ can be closely approximated with power laws in all considered cases.

**NLL with fixed temperature.** For all considered dataset–architecture pairs, and for all temperatures, $\widehat{\mathrm{NLL}}_n(\tau)$ with fixed $\tau$ can be closely approximated with a power law. Figure 1, middle and right shows an example approximation for VGG of the commonly used size with the temperature equal to one. Figure 2 shows the dynamics of the parameters $a$, $b$, $c$ of power laws, approximating the NLL with a fixed temperature of ensembles of different network sizes and for different temperatures, for VGG on a CIFAR-100 dataset. The rightmost plot reports the quality of approximation measured with RMSE in the *log*-space. We note that even the highest RMSE in the *log*-space corresponds to the low RMSE in the *linear* space (the RMSE in the *linear* space is less than 0.006 for all lines in figure 2).

In theory, starting from large enough $n$, $\mathrm{NLL}_n(\tau)$ follows a power law with parameter $a$ equal to -1, and for small $n$, more than one terms in eq. (2) are significant, resulting in a complex dependency $\mathrm{NLL}_n(\tau)$. In practice, we observe the power-law behaviour for the whole considered graph $\mathrm{NLL}_n(\tau)$, $n \geqslant 1$, but with $a$ slightly larger than $-1$. This result is consistent for all considered dataset–architecture pairs, see Appendix E.1.

When the temperature grows, the general behaviour is that $a$ approaches -1 more and more tightly. This behaviour breaks for the ensembles of small networks (blue lines). The reason is that the number of trained small networks is large, and the NLL for large ensembles with high temperature is noisy in log-scale, so the approximation of NLL with power law is slightly worse than for other settings, as confirmed in the rightmost plot of fig. 2. Nevertheless, these approximations are still very close to the data, we present the corresponding plots in Appendix E.2.

Parameter $b = \mathrm{NLL}_1(\tau) - \mathrm{NLL}_\infty(\tau)$ reflects the potential gain from the ensembling of the networks with the given network size and the given temperature. For a particular network size, the gain is higher for low temperatures, since networks with low temperatures are overconfident in their predictions, and ensembling reduces this overconfidence. With high temperatures, the predictions of both single network and ensemble get closer to the uniform distribution over classes, and $b$ approaches zero.

Parameter $c$ approximates the quality of the "infinite"-size ensemble, $\mathrm{NLL}_\infty(\tau)$. For each network size, there is an optimal temperature, which may be either higher or lower than one, depending on the dataset–architecture combination (see Appendix E.1 for more examples). This shows that even large ensembles need calibration. Moreover, the optimal temperature increases, when the network size grows. Therefore, not only large single networks are more confident in their predictions than small single networks, but the same holds even for large ensembles. Higher optimal temperature reduces their confidence. We notice that for given network size, the optimal temperature converges when $n \to \infty$, we show the corresponding plots in Appendix F.

**NLL with calibration.** When the temperature is applied before averaging, $\widehat{\mathrm{CNLL}}_n$ can be closely approximated with a power law for all considered dataset–architecture pairs, see Appendix G. Figure 3 shows how the resulting parameters of the power law change when the network size increases for different settings. The rightmost plot reports the quality of the approximation.

In figure 3, we observe that for WideResNet, parameter $b$ decreases, as $s$ becomes large, and $c$ starts growing for large $s$. For VGG, this effect also occurs in a light form but is almost invisible at the plot. This suggests that large networks gain less from the ensembling, and therefore the ensembles of larger networks are less effective than the ensembles of smaller networks. We suppose, the described

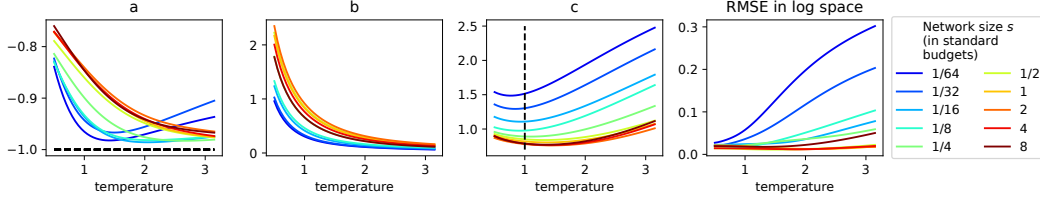

Figure 2: Parameters of power laws and the quality of the approximation for $\mathrm{NLL}_n(\tau)$ with a fixed temperature $\tau$ for VGG on CIFAR-100.

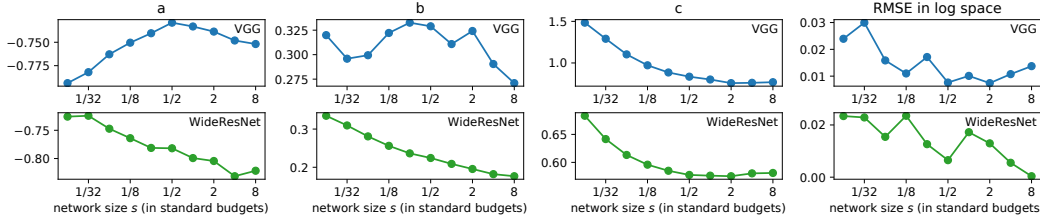

Figure 3: Parameters of power laws and the quality of the approximation for $\mathrm{CNLL}_n$ for different network sizes $s$. VGG and WideResNet on CIFAR-100.

effect is a consequence of under-regularization (the large networks need more careful hyperparameter tuning and regularization), because we also observed the described effect in a more strong form for the networks with all regularization turned off, see Appendix H. However, the described effect might also be a consequence of the decreased diversity of wider networks [23], and needs further investigation.

## 5 NLL as a function of network size

In this section, we analyze the behaviour of the NLL of the ensemble of a fixed size $n$ as a function of the member network size $s$. We consider both non-calibrated and calibrated NLL, and analyze separately cases $n = 1$ and $n > 1$. Geiger et al. [8] reason about a power law of accuracy for $n = 1$ when $s \to \infty$, considering shallow fully-connected networks on the MNIST dataset. We would like to check, whether $\widehat{(\mathrm{C})\mathrm{NLL}}_s$ can be approximated with a power law on the whole reasonable range of $s$ in practice.

**Single network.** Figure 4, left shows the NLL with $\tau = 1$ and the CNLL of a single VGG on the CIFAR-100 dataset as a function of the network size. We observe the double descent behaviour [22, 3] of the non-calibrated NLL, which could not be approximated with a power law for the considered range of $s$. The calibration removes the double-descent behaviour, and allows a close power-law approximation, as confirmed in the middle plot of figure 4. Interestingly, parameter $a$ is close to $-0.5$, which coincides with the results of Geiger et al. [8] derived for the test error. The results for other dataset–architecture pairs are given in Appendix I.

Nakkiran et al. [22] observe the double descent behaviour of *accuracy* as a function of the network size for highly overfitted networks, when training networks without regularization, with label noise, and for much more epochs than is usually needed in practice. In our practical setting, accuracy and CNLL are the monotonic functions of the network size, while for the non-calibrated NLL, the double descent behaviour is observed. Ashukha et al. [2] point out that accuracy and CNLL usually correlate, so we hypothesize that the double descent *may be* observed for CNLL in the same scenarios when it is observed for accuracy, while the non-calibrated NLL exhibits the double descent at the earlier epochs in these scenarios. To sum up, our results support the conclusions of [2] that the comparison of the NLL of the models of different *sizes* should only be performed with an optimal temperature.

**Ensemble.** As can be seen from figure 4, right, for ensemble sizes $n > 1$, CNLL starts increasing at some network size $s$. This agrees with the behaviour of parameter $c$ of the power law for CNLL

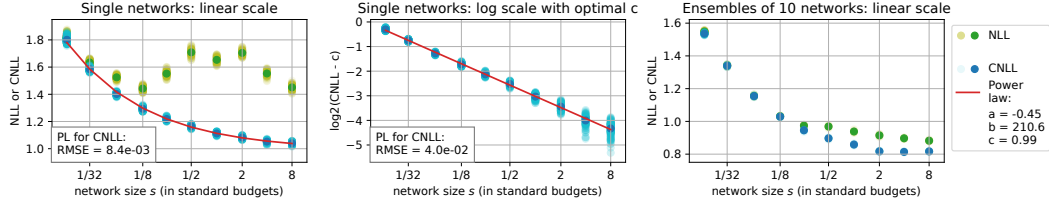

Figure 4: Non-calibrated $NLL_s$ and $CNLL_s$ for VGG on CIFAR-100. Left and middle: for a single network, $NLL_s$ exhibits double descent, while $CNLL_s$ can be closely approximated with a power law. Right: $NLL_s$ and $CNLL_s$ of an ensemble of several networks may be non-monotonic functions.

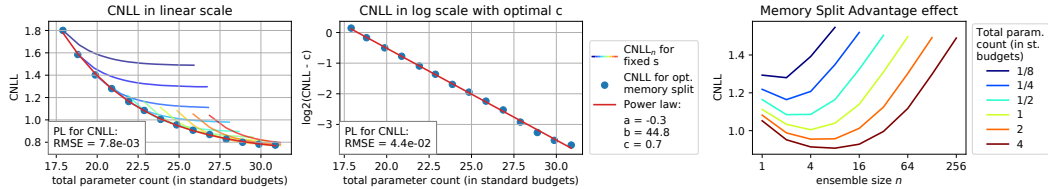

Figure 5: Left and middle: $CNLL_B$ for VGG on CIFAR-100 can be closely approximated with a power law. $CNLL_B$ is a lower envelope of $CNLL_n$ for different network sizes $s$. Right: Memory Split Advantage effect, VGG on CIFAR-100. For different memory budgets $B$, the optimal CNLL is achieved at $n > 1$.

shown in figure 3 and was discussed in section 4. Because of this behaviour, we do not perform experiments on approximating $CNLL_s$ for $n > 1$ with a power law.

# 6 NLL as a function of the total parameter count

In the previous sections, we analyzed the vertical and horizontal cuts of the $(n, s)$-plane shown in figure 1, left. In this section, we analyze the diagonal cuts of this space. One direction of diagonal corresponds to the fixed total parameter count, later referred to as a memory budget, and the orthogonal direction reflects the increasing budget.

We firstly investigate CNLL as a function of the memory budget. In figure 5, left, we plot sequences $\widehat{CNNL}_n$ for different network sizes $s$, aligning plots by the total parameter count. CNLL as a function of the memory budget is then introduced as the lower envelope of the described plots. As in the previous sections, we approximate this function with the power law, and observe, that the approximation is tight, the corresponding visualization is given in figure 5, middle. The same result for other dataset-architecture pairs is shown in Appendix J.

Another, practically important, effect is that the lower envelope may be reached at $n > 1$. In other words, for a fixed memory budget, a single network may perform worse than an ensemble of several medium-size networks of the same total parameter count, called a memory split in the subsequent discussion. We refer to the described effect itself as a Memory Split Advantage effect (MSA effect). We further illustrate the MSA effect in figure 5, right, where each line corresponds to a particular memory budget, the x-axis denotes the number of networks in the memory split, and the lowest CNLL, denoted at the y-axis, is achieved at $n > 1$ for all lines. We consistently observe the MSA effect for different settings and metrics, i.e. CNLL and accuracy, for a wide range of budgets, see Appendix J. We note that the MSA-effect holds even for budgets smaller than the standard budget. We also show in appendix J that using the memory split with the relatively small number of networks is only moderately slower than using a single wide network, in both training and testing stages. We describe the memory split advantage effect in more details in [4].

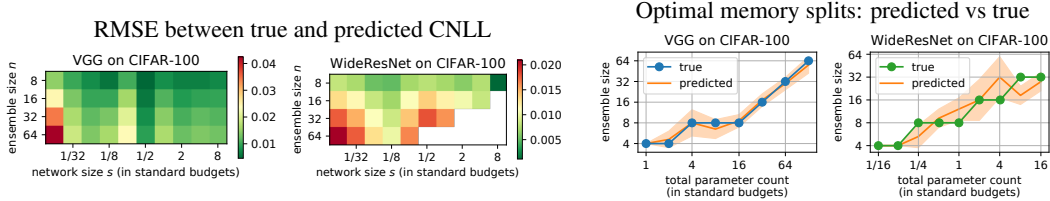

Figure 6: Predictions based on $\mathrm{CNLL}_n$ power laws for VGG and WideResNet on CIFAR-100. Predictions are made for large $n$ based on $n = 1..4$. Left pair: RMSE between true and predicted CNLL. Right pair: predicted optimal memory splits vs true ones. Mean $\pm$ standard deviation is shown for predictions.

# 7 Prediction based on power laws

One of the advantages of a power law is that, given a few starting points $y_1, \ldots, y_m$ satisfying the power law, one can exactly predict values $y_i$ for any $i \gg m$. In this section, we check whether the power laws discovered in section 4 are stable enough to allow accurate predictions.

We use the CNLL of the ensembles of sizes $1 - 4$ as starting points, and predict the CNLL of larger ensembles. We firstly conduct the experiment using the values of starting points, obtained by averaging over a large number of runs. In this case, the CNLL of large ensembles may be predicted with high precision, see Appendix K. Secondly, we conduct the experiment in the practical setting, when the values of starting points were obtained using only 6 trained networks (using 6 networks allows the more stable estimation of CNLL of ensembles of sizes $1 - 3$). The two left plots of figure 6 report the error of the prediction for the different ensemble sizes and network sizes of VGG and WideResNet on the CIFAR-100 dataset. The plots for other settings are given in Appendix K. The experiment was repeated 10 times for VGG and 5 times for WideResNet with the independent sets of networks, and we report the average error. The error is $1 - 2$ orders smaller than the value of CNLL, and based on this, we conclude that the discovered power laws allow quite accurate predictions.

In section 6, we introduced memory splitting, a simple yet effective method of improving the quality of the network, given the memory budget $B$. Using the obtained predictions for CNLL, we can now predict the optimal memory split (OMS) for a fixed $B$ by selecting the optimum at a specific diagonal of the predicted $(n, s)$-plane, see Appendix K for more details. We show the results for the practical setting with 6 given networks in figure 6, right. The plots depict the number of networks $n$ in the true and predicted OMS; the network size can be uniquely determined by $B$ and $n$. In most cases, the discovered power laws predict either the exact or the neighboring split. If we predict the neighboring split, the difference in CNLL between the true and predicted splits is negligible, i. e. of the same order as the errors presented in figure 6, left.

To sum up, we observe that the discovered power laws not only *interpolate* $\widehat{\mathrm{CNNL}}_n$ on the *whole* considered range of $n$, but also is able to *extrapolate* this sequence, i. e. CNLL fitted on a *short* segment of $n$ approximates well the *full* range, providing an argument for using particularly power laws and not other functions.

# 8 Related Work

**Deep ensembles and overparameterization.** The two main approaches to improve deep neural networks accuracy are ensembling and increasing network size. While a bunch of works report the quantitative influence of the above-mentioned techniques on model quality [7, 12, 18, 22, 24, 25], few investigate the qualitative side of the effect. Some recent works [5, 8, 9] consider a simplified or narrowed setup to tackle it. For instance, Geiger et al. [8] similarly discover the power laws in test error w. r. t. model and ensemble size for simple binary classification with hinge loss, and give a heuristic argument supporting their findings. We provide an extensive theoretical and empirical justification of similar claims for the calibrated NLL using modern architectures and datasets. Other layers of works on studying neural network ensembles and overparameterized models include but not limited to the

Bayesian perspective [10, 31, 32], ensembles diversity improvement techniques [14, 20, 28, 34], neural tangent kernel (NTK) view on overparameterized neural networks [1, 11, 19], etc.

**Power laws for predictions.**    A few recent works also empirically discover power laws with respect to data and model size and use them to extrapolate the performance on small models/datasets to larger scales [13, 26]. Their findings even allow estimating the optimal compute budget allocation given limited resources. However, these studies do not account for the ensembling of models and the calibration of NLL.

**MSA-effect.**    Concurrently with our work, Kondratyuk et al. [15] investigate a similar effect for budgets measured in FLOPs. Earlier, an MSA-like effect has also been noted in [6, 21]. However, the mentioned works did not consider the proper regularization of networks of different sizes and did not propose the method for predicting the OMS, while both aspects are important in practice.

## 9    Conclusion

In this work, we investigated the power-law behaviour of CNLL of deep ensembles as a function of ensemble size $n$ and network size $s$ and observed the following power laws. Firstly, with a minor modification of the calibration procedure, CNLL as a function of $n$ follows a power law on the wide finite range of $n$, starting from $n = 1$, but with the power parameter slightly higher than the one derived theoretically. Secondly, the CNLL of a single network follows a power law as a function of the network size $s$ on the whole reasonable range of network sizes, with the power parameter approximately the same as derived. Thirdly, the CNLL also follows a power law as a function of the total parameter count (memory budget). The discovered power laws allow predicting the quality of large ensembles based on the quality of the smaller ensembles consisting of networks with the same architecture. The practically important finding is that for a given memory budget, the number of networks in the optimal memory split is usually much higher than one, and can be predicted using the discovered power laws. Our source code is available at `https://github.com/nadiinchi/power_laws_deep_ensembles`.

## Broader Impact

In this work, we provide an empirical and theoretical study of existing models (namely, deep ensembles); we propose neither new technologies nor architectures, thus we are not aware of its specific ethical or future societal impact. We, however, would like to point out a few benefits gained from our findings, such as optimization of resource consumption when training neural networks and contribution to the overall understanding of neural models. As far as we are concerned, no negative consequences may follow from our research.

## Acknowledgments and Disclosure of Funding

We would like to thank Dmitry Molchanov, Arsenii Ashukha, and Kirill Struminsky for the valuable feedback. The theoretical results presented in section 2 were supported by Samsung Research, Samsung Electronics. The empirical results presented in sections 4, 5, 6, 7 were supported by the Russian Science Foundation grant №19-71-30020. This research was supported in part through the computational resources of HPC facilities at NRU HSE. Additional revenues of the authors for the last three years: Stipend by Lomonosov Moscow State University, Travel support by ICML, NeurIPS, Google, NTNU, DESY, UCM.

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
