[Supplementary Material]

# A Theory

## A.1 Proof of Proposition 1

In this section, we prove Proposition 1 about the asymptotic power law behavior of the model-average NLL of the ensemble for a single object, $\text{NLL}_n^{\text{obj}}$, as a function of the ensemble size $n$.

**Proof.** In what follows, by abuse of notation, the subscript $\text{obj}$ is omitted. Consider $n$ independent identically distributed random variables $p_i^* \in [\epsilon, 1]$, $\epsilon > 0$ with mean $\mu = \mathbb{E}p_i^*$ and variance $\sigma^2 = \mathbb{D}p_i^*$. Denote $\bar{p}_n^* = \frac{1}{n}\sum_{i=1}^n p_i^*$. The cornerstone of the proof is doing Taylor expansion of logarithm around $\mu$ and obtaining the following expression:

$$\text{NLL}_n^{\text{obj}} = -\mathbb{E}\log(\bar{p}_n^*) = \mathbb{E}\left[-\log(\mu) + \frac{\mu - \bar{p}_n^*}{\mu} + \frac{(\mu - \bar{p}_n^*)^2}{2\mu^2} + R_2(\mu - \bar{p}_n^*)\right] =$$

$$= -\log(\mu) + \frac{1}{n}\frac{\sigma^2}{2\mu^2} + \mathbb{E}\left[R_2(\mu - \bar{p}_n^*)\right], \quad (7)$$

where $R_m(\mu - \bar{p}_n^*) = -\log\left(\frac{\bar{p}_n^*}{\mu}\right) - \sum_{k=1}^m \frac{(\mu - \bar{p}_n^*)^k}{k\mu^k}$ is the remainder term. First we show how to bound $\mathbb{E}\left[R_m(\mu - \bar{p}_n^*)\right]$ w.r.t. $n$ for general $m$ and then use this bound to derive the $\mathcal{O}\left(\frac{1}{n^2}\right)$ asymptotic for $\mathbb{E}\left[R_2(\mu - \bar{p}_n^*)\right]$.

Let us fix some $0 < \varepsilon < \mu$ and split the expectation of the remainder into two terms:

$$\mathbb{E}\left[R_m(\mu - \bar{p}_n^*)\right] = \mathbb{E}\left[R_m(\mu - \bar{p}_n^*) \cdot \mathbb{1}[|\bar{p}_n^* - \mu| > \varepsilon]\right] + \mathbb{E}\left[R_m(\mu - \bar{p}_n^*) \cdot \mathbb{1}[|\bar{p}_n^* - \mu| \leq \varepsilon]\right]. \quad (8)$$

Applying the Hoeffding's inequality to $\bar{p}_n^*$ allows us to exponentially bound the probability of deviation from $\mu$ by $\varepsilon$:

$$P(|\bar{p}_n^* - \mu| > \varepsilon) \leq 2\exp\left(-2n\varepsilon^2\right). \quad (9)$$

Taking inequality (9) into account, we can bound the first term in (8) as follows:

$$\left|\mathbb{E}\left[R_m(\mu - \bar{p}_n^*) \cdot \mathbb{1}[|\bar{p}_n^* - \mu| > \varepsilon]\right]\right| \leq 2M\exp\left(-2n\varepsilon^2\right), \quad (10)$$

where $M = \max_{\bar{p}_n^* \in [\epsilon, 1]} |R_m(\mu - \bar{p}_n^*)|$.

In the second term of (8), as $|\bar{p}_n^* - \mu| \leq \varepsilon < \mu$, the remainder term can be expanded as a Taylor series $R_m(\mu - \bar{p}_n^*) = \sum_{k=m+1}^\infty \frac{(\mu - \bar{p}_n^*)^k}{k\mu^k}$ and thus bounded as follows:

$$|R_m(\mu - \bar{p}_n^*)| \leq \sum_{k=m+1}^\infty \frac{|\mu - \bar{p}_n^*|^k}{k\mu^k} \leq \sum_{k=m+1}^\infty \frac{\varepsilon^k}{k\mu^k}. \quad (11)$$

From that we derive the following bound on the second term in (8):

$$\left|\mathbb{E}\left[R_m(\mu - \bar{p}_n^*) \cdot \mathbb{1}[|\bar{p}_n^* - \mu| \leq \varepsilon]\right]\right| \leq \sum_{k=m+1}^\infty \frac{\varepsilon^k}{k\mu^k} = \mathcal{O}\left(\varepsilon^{m+1}\right). \quad (12)$$

If we fix any $0 < \delta < 1$ and choose the following sequence of $\varepsilon_n = n^{-\frac{1-\delta}{2}}$,[1] the first bound (10) becomes $2M\exp\left(-2n^\delta\right)$, i.e. exponential in $n$, and the second bound (12) becomes $\mathcal{O}\left(n^{-\frac{(m+1)(1-\delta)}{2}}\right)$. By taking $\delta = \frac{1}{5}$, we obtain that $\mathbb{E}\left[R_4(\mu - \bar{p}_n^*)\right] = \mathcal{O}\left(\frac{1}{n^2}\right)$.

To conclude the proof, we note that

$$\mathbb{E}\left[R_2(\mu - \bar{p}_n^*)\right] = \frac{\mathbb{E}(\mu - \bar{p}_n^*)^3}{3\mu^3} + \frac{\mathbb{E}(\mu - \bar{p}_n^*)^4}{4\mu^4} + \mathbb{E}\left[R_4(\mu - \bar{p}_n^*)\right] \quad (13)$$

and $\mathbb{E}(\mu - \bar{p}_n^*)^3 = \frac{\mathbb{E}(\mu - p_i^*)^3}{n^2} = \mathcal{O}\left(\frac{1}{n^2}\right)$, $\mathbb{E}(\mu - \bar{p}_n^*)^4 = \frac{3\sigma^4}{n^2} + \mathcal{O}\left(\frac{1}{n^3}\right) = \mathcal{O}\left(\frac{1}{n^2}\right)$. ∎

## A.2 Power law in NLL with temperature applied after averaging

Here we consider the power-law behaviour of the calibrated NLL of the ensemble, when the temperature is applied after averaging. Recall that $K$ is the number of classes and operator $*$ denotes retrieving the prediction for the correct class. Denote $\bar{p}_n = \frac{1}{n}\sum_{i=1}^n p_i$, $\mu = \mathbb{E}p_i$. Then the single-object model-average NLL has the following form (suppose, without loss of generality, that the first class is the correct one):

$$\text{NLL}_n^{\text{obj}}(\tau) = -\mathbb{E}\log\left(\text{softmax}\{(\log(\bar{p}_n))/\tau\}^*\right) = \mathbb{E}\left[-\gamma\log(\bar{p}_{n1}) + \log\left(\sum_{k=1}^K \bar{p}_{nk}^\gamma\right)\right], \quad (14)$$

where $\gamma = \frac{1}{\tau}$. By applying a similar Taylor expansions trick as in Proposition 1 to the function $f(p) = -\gamma \log(p_1) + \log \left( \sum_{k=1}^{K} p_k^{\gamma} \right)$ under expectation in (14), we arrive at:

$$\text{NLL}_n^{\text{obj}}(\tau) \approx f(\mu) + \frac{1}{2n} \sum_{k,k'=1}^{K} cov(p_{ik}, p_{ik'}) \left. \frac{\partial^2 f}{\partial p_k \partial p_{k'}} \right|_{p=\mu}, \tag{15}$$

where

$$\frac{\partial^2 f}{\partial p_k \partial p_{k'}} = \frac{\gamma}{p_1^2}[k=1][k'=1] - \frac{\gamma^2 p_k^{\gamma-1} p_{k'}^{\gamma-1}}{\left( \sum_{s=1}^{K} p_s^{\gamma} \right)^2} + [k=k']\frac{\gamma(\gamma-1)p_k^{\gamma-2}}{\sum_{s=1}^{K} p_s^{\gamma}}. \tag{16}$$

We can see that the first term in (16) is non-negative, while the second one is, on the contrary, always negative. High values of temperature (i. e. $\gamma < 1$) make the last term negative as well, which, in turn, may lead to negative $b$ coefficient in the power law of the right-hand part of (15). We observe this effect in practice: at certain values of temperature applied after averaging, the NLL starts *increasing* as a function of $n$, see Appendix C.2. We could not apply a similar technique, as was provided in section A.1, to derive the exact asymptote in (15), when the temperature is applied after averaging.

We note, however, that when the temperature $\tau$ is applied *before* averaging, i. e. fixed[2] $\tau$ is used to compute the final member networks predictions: $p_i(\tau) = \text{softmax}\{\log(p_i)/\tau\}$, the expression for $\text{NLL}_n^{\text{obj}}(\tau)$ fits the form of (1):

$$\text{NLL}_n^{\text{obj}}(\tau) = -\mathbb{E} \log \left( \frac{1}{n} \sum_{i=1}^{n} p_i^*(\tau) \right), \tag{17}$$

and hence Proposition 1 is applicable. The power law parameter $b = \frac{\sigma^2}{2\mu^2}$ in this case is always positive.

### A.3 Lower envelope of power laws with continuous temperature

In this subsection, we show that even when the set of temperatures is uncountable, the lower envelope of power laws asymptotically follows the power law. This argument generalizes our discussion at the end of section 2.

**Proposition 2** *Consider a compact set $\mathcal{T}$ and two continuous mappings $c : \mathcal{T} \to \mathbb{R}$ and $b : \mathcal{T} \to \mathbb{R}$. Let each value $\tau \in \mathcal{T}$ correspond to a certain power law w. r. t. $n$:*

$$\text{PL}_n(\tau) = c(\tau) + \frac{b(\tau)}{n}. \tag{18}$$

*Then the lower envelope $\text{LE}_n = \min_{\tau \in \mathcal{T}} \text{PL}_n(\tau)$ of the power laws (18) follows a power law asymptotically.*

**Proof.** Let $\tau_n \in \underset{\tau \in \mathcal{T}}{\text{Argmin}} \, \text{PL}_n(\tau)$ be the value which minimizes $\text{PL}_n(\tau)$ at given $n$. Then the lower envelope of power laws (18) can be defined as

$$\text{LE}_n = \text{PL}_n(\tau_n). \tag{19}$$

We need to show that $\exists c^*, b^* \in \mathbb{R}$:

$$\text{LE}_n = c^* + \frac{b^*}{n} + o\left( \frac{1}{n} \right). \tag{20}$$

By the definition of $\tau_n$, the following inequalities hold:

$$\begin{cases} c(\tau_n) + \frac{b(\tau_n)}{n} \leq c(\tau_{n+1}) + \frac{b(\tau_{n+1})}{n} \\ c(\tau_{n+1}) + \frac{b(\tau_{n+1})}{n+1} \leq c(\tau_n) + \frac{b(\tau_n)}{n+1} \end{cases} \implies \begin{cases} c(\tau_{n+1}) \leq c(\tau_n) \\ b(\tau_n) \leq b(\tau_{n+1}). \end{cases} \tag{21}$$

The first of the right inequalities in (21) can be obtained via multiplication of the left inequalities by $n$ and $n+1$, respectively, and summation. The second inequality is obtained after substituting the first one into the right-hand part of the first left inequality.

Now, as $\{\tau_n\} \subset \mathcal{T}$, which is a compact, there exists a subsequence $\tau_{n_k} \to \tau^* \in \mathcal{T}$ which implies that $b(\tau_n) \to b(\tau^*)$, $c(\tau_n) \to c(\tau^*)$ monotonically due to continuity of mappings $b(\tau)$, $c(\tau)$ and the right inequalities in (21). Finally, consider the difference between $\text{LE}_n$ and $\text{PL}_n(\tau^*)$:

$$0 \leq \text{PL}_n(\tau^*) - \text{LE}_n = c(\tau^*) - c(\tau_n) + \frac{b(\tau^*) - b(\tau_n)}{n} \leq \frac{b(\tau^*) - b(\tau_n)}{n}. \tag{22}$$

As $b(\tau^*) - b(\tau_n) = o(1)$, we come to (20) with $c^* = c(\tau^*)$, $b^* = b(\tau^*)$. ∎

As was shown in section 2, $\text{NLL}_n(\tau)$ follows a power law asymptotically for any fixed $\tau > 0$. Due to continuity of a softmax function w. r. t. $\tau$, we could deduce that the parameters $b$ and $c$ of the respective power law are also continuous functions of temperature. Finally, appropriately choosing the temperatures set $\mathcal{T}$ (namely, separate from zero — the singularity point) allows to conclude that deviation of $\text{NLL}_n(\tau)$ from its power law is $o(\frac{1}{n})$ uniformly for all $\tau \in \mathcal{T}$ and hence Proposition 2 is applicable to the lower envelope of $\text{NLL}_n(\tau)$ as well.

# B Experimental details

**Data.** We conduct experiments on CIFAR-100 and CIFAR-10 datasets, each containing 50000 training and 10000 testing examples. For tuning hyperparameters, we randomly select 5000 training examples as a validation set. After choosing optimal hyperparameters, we retrain the models on the full training dataset. We use a standard data augmentation scheme: zero-padding with 4 pixels on each side, random cropping to produce $32 \times 32$ images, and horizontal mirroring with probability 0.5.

**Details.** We consider two architectures: VGG and WideResNet. We use the implementation provided at `https://github.com/timgaripov/dnn-mode-connectivity`.To obtain networks of different sizes, we vary the width factor of the networks: for VGG/WideResNet, we use convolutional layers with $[w, 2w, 4w, 8w] / [w, 2w, 4w]$ filters, and fully-connected layers with $8w/4w$ neurons. For VGG/WideResNet, we consider $2 \leqslant w \leqslant 181 / 5 \leqslant w \leqslant 453$; $w = 64/160$ corresponds to a standard, commonly used, configuration with $s_{\text{standard}} = 15.3\text{M} / 36.8\text{M}$ parameters. These sizes are referred to as the standard budgets. For VGG, we use weight decay, and binary dropout for fully-connected layers. For WideResNet, we use weight decay and batch normalization. For each considered width factor, we tune the hyperparameters using grid search with the following grids: learning rate — $\{0.005, 0.05\} / \{0.01, 0.1\}$ for VGG/WideResNet, weight decay — $\{10^{-4}, 3 \cdot 10^{-4}, 10^{-3}, 3 \cdot 10^{-3}\}$, dropout rate — $\{0, 0.25, 0.5\}$. We train all models for 200 epochs using SGD with momentum of 0.9 and the following learning rate schedule: constant (100 epochs) – linearly annealing (80 epochs) – constant (20 epochs). The final learning rate is 100 times smaller than the initial one. We use the batch size of 128.

In several experiments presented in the Appendix, we train networks without regularization, with all hyperparameters being the same for all network sizes. By this, we mean that we set weight decay and dropout rate to zero, do not use data augmentation, and use an initial learning rate 10 times smaller than in the reference implementation to ensure that the training converges for all considered models.

**Computing infrastructure.** VGG networks were trained on NVIDIA Teals P100 GPU. Training one network of the standard size / smallest considered size / largest considered size took approximately 1 hour / 20 minutes / 4.5 hours. WideResNet networks were trained on NVIDIA Tesla V100 GPU. Training one network of the standard size / smallest considered size / largest considered size took approximately 5.5 hours / 50 minutes / 32 hours.

**Approximating sequences with power laws.** To approximate a sequence $\hat{y}_m$, $m \geqslant 1$ with a power law, we solve optimization task (6) using BFGS and performing computations with the double-precision floating point numbers (float64) to avoid overflow when computing the logarithms of nearly zero values. If we consider a uniform grid for $m$ in the linear scale, then in the logarithmic scale the density of the points in the area with large $m$ is much higher than in the area with small $m$. To account for this variable density, we weight terms corresponding to different values of $m$ in (6): $w_m = \frac{1}{m}$. When we report the quality of approximation measured with RMSE, we use uniform weights $w_m = \frac{1}{M}$. The described heuristic is utilized for approximating $\widehat{(\text{C})\text{NLL}}_n$, since we have a uniform grid for $n$ in the linear scale, and is not utilized for approximating $\widehat{\text{CNLL}}_s$ and $\widehat{\text{CNLL}}_B$, since for these sequences we have a uniform grid for $s/B$ in the logarithmic scale.

**Test-time cross-validation.** Ashukha et al. [2] utilize a so-called *test-time cross-validation* to obtain an unbiased, low-variance estimate of the CNLL using the publicly available test set. The test-time cross-validation implies that the test set is randomly split into two equal parts five times. For each split, one part is used to find the optimal temperature, and the other one — to measure the CNLL, and vice versa. Finally, the CNLL is averaged over ten measurements.

# C Calibration of ensembles: applying temperature before or after averaging

## C.1 Difference in CNLL

In section 2, we introduced two ways of applying temperature to the ensemble, namely before and after averaging. In this subsection, we empirically compare these two calibration procedures. Figure 7 shows the results for VGG on CIFAR-100, for setting with and without regularization. The difference between CNLL values is low in all the cases, hence the two procedures perform similarly. In most of the cases, particularly in practically important cases of ensembling networks of medium and large sizes, calibration with applying temperature before averaging performs slightly better (there are a lot of green pixels in the heatmaps). The results for other dataset-architecture pairs are similar. To sum up, the procedure with applying temperature before averaging can be used in practice instead of the standard one, with temperature applied after averaging, without loss in the quality.

Figure 7: Difference between CNLLs computed with applying temperature *before* and *after* averaging for different values of ensemble size $n$ and network size $s$, for VGG on CIFAR-100. Left: the optimal temperature for CNLL is chosen using the whole test set. Right: CNLL is computed using test-time cross-validation.

Figure 8: $\text{NLL}_n(\tau)$ for different values of $\tau$ and $\text{CNLL}_n$. VGG on CIFAR-100, the smallest considered network size (1/64 of standard budget). The blue / red color corresponds to the low / high temperatures.

## C.2 Dynamics of NLL with fixed temperature and CNLL

In this subsection, we illustrate the behaviour of $\text{NLL}_n(\tau)$ and $\text{CNLL}_n$ for both ways of applying temperature. Figure 8 shows the results for VGG on CIFAR-100, for setting with and without regularization, for the smallest considered network size. When the temperature is applied before averaging, in practice, both $\text{NLL}_n(\tau)$ and $\text{CNLL}_n$ follow a power law as functions of $n$, with positive parameter $b$, as can be seen from figure 8, left column. When the temperature is applied after averaging, $\text{NLL}_n(\tau)$ starts growing w.r.t. $n$ for high values of $\tau$, as can be seen from figure 8, right. As a result, $\text{CNLL}_n$ in this case may be non-convex, see the bottom right plot. We observed the effect of $\text{NLL}_n(\tau)$ increase in a wide range of settings, while the effect of $\text{CNLL}_n$ non-convexity was observed only for small unregularized networks. In the most cases, $\text{CNLL}_n$ with temperature applied after averaging also follows a power law.

## D Comparison of CNLL and LE-NLL

In this section, we empirically show that moving the minimum operation outside the expectation in equation (3) does not change the value of CNLL a lot in practice. In other words, we compare the values of CNLL (3), commonly used in practice, and LE-NLL (4), utilized in section 2, for different dataset–architecture pairs. We

Figure 9: Difference between CNLL (3) and LE-NLL (4) for different values of ensemble size $n$ and network size $s$, for VGG on CIFAR-100. Left: the optimal temperature for CNLL is chosen using the whole test set. Right: CNLL is computed using test-time cross-validation.

Figure 10: Parameters of power laws and the quality of approximation for $\mathrm{NLL}_n(\tau)$ with fixed temperature $\tau$.

consider two scenarios for computing CNLL: (a) when the optimal temperature is chosen using the whole test set, as it is the case for LE-NLL, and (b) when the test-time cross-validation is utilized [2]. In figure 9 we depict the difference between CNLL and LE-NLL for different values of ensemble size $n$ and network size $s$, for VGG on CIFAR-100. We observe that the difference is negligible, compared to the values of LE-NLL. The relative difference for all values of $n$ and $s$ is bounded by $0.018\%/0.29\%$ for scenarios (a) and (b) respectively. For WideResNet on CIFAR-100, the relative difference is bounded by $0.081\%/0.61\%$, for VGG on CIFAR-10 — $0.054\%/0.46\%$, for WideResNet on CIFAR-10 — $0.023\%/0.45\%$ for scenarios (a) and (b) respectively. In all the the experiments in the paper, we use CNLL computed using test-time cross-validation.

VGG on CIFAR-100: temperature $\tau = 1$      VGG on CIFAR-100: temperature $\tau = 3$

Figure 11: Power law approximation for $\text{NLL}_n(\tau)$ for VGG of small size (1/64 of standard budget) on CIFAR-100, with the standard and high values of temperature. Approximations in both linear space and log-space are presented.

# E   NLL with fixed temperature as a function of ensemble size

## E.1   Power law approximation for different dataset–architecture pairs

In this subsection, we show that $\text{NLL}_n(\tau)$ can be closely approximated with a power law as a function of $n$, on the whole considered range $n \geqslant 1$, for all values of $\tau > 0$, and for different dataset–architecture pairs. Figure 10 supplements figure 2 for other dataset–architecture pairs, and shows the quality of approximation of $\text{NLL}_n(\tau)$ with power laws as well as the dynamics of power law parameters $a$, $b$, $c$.

For a particular network size, parameter $a$ is generally a decreasing function of temperature $\tau$. The described effect does not hold for the ensembles of large WideResNet networks, because for them we only consider small ensembles due to the resource limitations, so the approximation is performed for only a few points and is slightly unstable. The described effect also does not hold for the ensembles of small VGG networks, since applying high temperatures leads to the noise in $\text{NLL}_n(\tau)$ for large $n$ and makes the approximation slightly worse than for other settings, as confirmed in the rightmost plot. This approximation is still very close, see appendix E.2.

Parameter $b$ decreases when the temperature grows, since lower temperatures result in more contrast predictions, and ensembling smooths them. Parameter $c$ is a non-monotonic function of $\tau$, and its optimum reflects the optimal temperature for the "infinite"-size ensemble. The optimal temperature may be greater or less than one, depending on the dataset–architecture combination.

## E.2   Power law approximation for small networks with high temperature

Figure 11 visualizes the power law approximation of $\text{NLL}_n(\tau)$ for VGG of the smallest considered width, with a standard temperature $\tau = 1$ and a high temperature $\tau = 3$. In log-space, we observe the visible fluctuations of $\text{NLL}_n(\tau = 3)$ for large ensembles. These fluctuations result in a small bias in the estimation of power law parameters, particularly parameter $a$. However, in the linear space, the fluctuations do not have a significant effect, and the approximation is close to the data.

The reason for the fluctuations is that for high values of temperature and for large ensemble sizes $n$, the difference $\text{NLL}_n(\tau) - c$ for optimal $c$ is extremely small, while the precision of the computation is limited. Particularly, the minimum value of $\log_2(\text{NLL}_n(\tau) - c)$ is approximately equal to $-11$ for $\tau = 3$ and to $-9$ for $\tau = 1$, see the y-axes of figure 11. We observe the fluctuations for the ensembles of small networks, since for small $s$ we consider the largest values of $n$.

# F   Convergence of temperature

To show that the optimal temperature $\tau$ converges when ensemble size $n$ increases, we plot optimal $\tau$ vs. $n$ for different dataset–architecture pairs in figure 12. We average the optimal temperature over runs, i. e. different trained ensembles, and over folds in test-time cross-validation.

# G   Power law approximation of CNLL as a function of ensemble size

Figure 13 supplements figure 3, to show that $\text{CNLL}_n$ with the temperature applied before averaging can be closely approximated with a power law on the whole considered range $n \geqslant 1$, for all considered dataset–architecture pairs. The rightmost plot reports the quality of approximation in the log-space. We notice that in the linear space, the RMSE is less than $5.9 \cdot 10^{-4}$ for all blue points, corresponding to VGG, and less $4.7 \cdot 10^{-4}$ for

Figure 12: Optimal temperatures for different dataset–architecture pairs.

Figure 13: Parameters of power laws and the quality of approximation for $\mathrm{CNLL}_n$ for different network sizes $s$. VGG and WideResNet on CIFAR-10.

all green points, corresponding to WideResNet. The first three plots in each row visualize the dynamics of power law parameters $a$, $b$, $c$ as network size increases; these dynamics were discussed in section 4.

# H    CNLL of the ensemble of unregularized networks

In this section, we analyse the CNLL of the ensembles of unregularized networks as a function of network size. Turning off regularization, i. e. weight decay and dropout, results in the clearly observed effect that for large values of $n$, $\mathrm{CNLL}_s$ of the ensemble of size $n$ has optimum at some network size $s$. We firstly visualize this effect in figure 14 showing the $(n, s)$ plane of CNLL for the setting without regularization, for VGG and WideResNet on CIFAR-100. For higher $n$ (e. g. $n > 5$), the horizontal cuts are non-monotonic w. r. t. $s$. Secondly, we analyse the CNLL of the "infinite"-size ensemble, using power laws discovered in the paper. Figure 15 visualizes the parameters of the power laws approximating $\mathrm{CNLL}_n$ for different values of $s$, for VGG and WideresNet on CIFAR-100. We can clearly observe that parameter $b$, that reflects the possible gain of ensembling the networks of size $s$, decreases as $s$ increases, while the parameter $c$, that reflects the CNLL of the "infinite"-size ensemble, is non-monotonic, i. e. achieves optimum at some $s$ in the middle of the considered range of $s$. The similar effect was observed in figure 3 for the regularized setting, but in a less visible form. We notice that the decrease of parameter $b$ in the setting without regularization is much larger, than in the standard setting with regularization, and the effect of non-monotonicity of $c$ is also stronger.

The described effect of the non-monotonicity of $\mathrm{CNLL}_s$ for large ensembles may be a consequence of under-regularization of the large networks, or a consequence of the decreased diversity of the large networks [23], and needs further investigation.

Figure 14: The $(n, s)$-plane of CNLL for WideResNet and VGG on CIFAR-100 for the setting without regularization.

Figure 15: Parameters of power laws and the quality of approximation for $\text{CNLL}_n$ for different network sizes $s$. VGG and WideResNet on CIFAR-100, setting without regularization.

Figure 16: Non-calibrated $\text{NLL}_s$ and $\text{CNLL}_s$. Left and middle: for a single network, $\text{NLL}_s$ may exhibit double descent, while $\text{CNLL}_s$ can be closely approximated with a power law. Right: $\text{NLL}_s$ and $\text{CNLL}_s$ of an ensemble of several networks may be non-monotonic functions.

## I   Power law approximation of NLL as a function of network size

Figure 16 supplements figure 4, to show that $\text{CNLL}_s$ can be closely approximated with a power law on the whole considered range of network sizes $s$, for all considered dataset–architecture pairs, with parameter $a$ close to $-0.5$. For the non-calibrated NLL of VGG on CIFAR-10, we observe the same double descend behaviour as for VGG on CIFAR-100.

## J   Power law approximation of NLL as a function of the memory budget. MSA effect

Figure 17 supplements figure 5, to show that $\text{CNLL}_B$ can be closely approximated with a power law on the whole considered range of memory budgets $B$, for all considered dataset–architecture pairs. The right column of the plots visualizes the MSA effect: for each memory budget $B$ (each line), the optimum of CNLL is achieved at abscissa $n > 1$. The MSA effect also holds for accuracy for a wide range of budgets, including budgets less than the standard one, see figure 18. The optimal memory split is usually achieved for accuracy at the same $n$ or at the smaller $n$ than for CNLL.

## WideResNet on CIFAR-100

## VGG on CIFAR-10

## WideResNet on CIFAR-10

Figure 17: Left and middle: $\text{CNLL}_B$ for different dataset–architecture pairs can be closely approximated with a power law. $\text{CNLL}_B$ is a lower envelope of $\text{CNLL}_n$ for different network sizes $s$. Right: MSA effect: for different memory budgets $B$, the optimal CNLL is achieved at $n > 1$.

## VGG on CIFAR-100            ## WideResNet on CIFAR-100

## VGG on CIFAR-10            ## WideResNet on CIFAR-10

Figure 18: MSA effect for accuracy: for different memory budgets $B$, the optimal accuracy is achieved at $n > 1$.

Figure 19: Predictions based on $\text{CNLL}_n$ power laws. Difference between true and predicted CNLL. Predictions are made for large $n$ based on $n = 1..4$ using all trained networks of each size, i.e. with averaging $\text{CNLL}_n$ for $n = 1..4$ over a large number of runs.

Figure 20: Predictions based on $\text{CNLL}_n$ power laws for VGG and WideResNet on CIFAR-10. Predictions are made for large $n$ based on $n = 1..4$ using 6 trained networks of each size. Left pair: RMSE between true and predicted CNLL. Right pair: predicted optimal memory splits vs true ones. Mean $\pm$ standard deviation is shown for predictions.

**The timing analysis of the memory splitting procedure.** One might wonder, how much slower is training and prediction with the ensemble of several medium-size networks compared to a single large network. In table 1, we list the training and prediction time for a single VGG on CIFAR-100, for different network sizes. We conduct this experiment on GPU conducting training / prediction with several networks sequentially, using the training batch size of 64 and the testing batch size of 1024. We observe that using the memory split with the relatively small number of networks (which is the case in practice) is only moderately slower than using a single wide network. For example, for budget $4S$, with a single network / memory split of 4 networks, testing takes 111 / 132 sec, while one training epoch takes 42 / 64 seconds. Please note that these numbers are given for the case when the networks in the memory split are trained / validated *sequentially*, while the memory split allows *parallel* training / validation.

| Network size (in standard budgets) | 0.125 | 0.25 | 0.5 | 1 | 2 | 4 | 8 |
|---|---|---|---|---|---|---|---|
| Time of 1 training epoch | 7.8 | 8 | 11 | 16 | 27 | 42 | 80 |
| Prediction time | 6.8 | 9.5 | 20 | 33 | 63 | 111 | 227 |

Table 1: Training and prediction time for a single VGG of different sizes on CIFAR-100.

# K    Predictions based on power laws

In this section, we expand the discussion on using the power laws observed in the paper for predicting the CNLL of large ensembles, and optimal memory splits for different memory budgets. For all predictions, we use the values of $\text{CNLL}_n$ for $n = 1..4$ for different values of $s$ as given data.

We firstly conduct an experiment in which $\text{CNLL}_n$ for $n = 1..4$ is averaged over all available $\lfloor \frac{\ell}{n} \rfloor$ runs, see section 3 for details on the number of available runs. In this experiment, we predict $\text{CNLL}_n$ for $n > 4$. Figure 19 reports the RMSE between the true and predicted $\text{CNLL}_n$. The predictions are highly precise with the error 2–3 orders smaller than the values being predicted. This experiment provides an evidence that the power laws discussed in the paper can be used for prediction. However, the described setting is not practically applicable due to the use of a large number of networks.

The second experiment is practically oriented: we use only 6 networks of each network size $s$, and average $\text{CNLL}_n$ for $n = 1..4$ using only these 6 networks. The number 6 is chosen to provide more stable CNLL estimates for $n = 1..3$. We average the errors in predictions over 10 / 5 runs for VGG / WideResNet. Figure 20 supplements figure 6 and shows the results for CIFAR-10. When predicting $\text{CNLL}_n$ for $n > 4$ (see the left pairs of plots in figures 6 and 20), we obtain the error of 1-2 orders smaller than the values being predicted. The

**Algorithm 1** Finding optimal memory split using power laws
___
**Input:** budget $B$ (the number of parameters);
**Output:** the optimal memory split of budget $B$;
 1: $\text{CNLL}_* = +\infty$, $n_* = \text{None}$; $k = -1$;
 2: **repeat**
 3:     $k \mathrel{+}= 1, n = 2^k$;
 4:     Train $\min(n, 6)$ networks of size $B/n$;
 5:     **if** $n > 4$ **then**
 6:         Fit the parameters of power law on sequence $\{\widehat{\text{CNLL}}_i\}_{i=1}^4$;
 7:         Predict $\text{CNLL}_n$ using power law;
 8:     **else**
 9:         Compute $\text{CNLL}_n$ using trained networks;
10:     **if** $\text{CNLL}_n < \text{CNLL}_*$ **then**
11:         $\text{CNLL}_* = \text{CNLL}_n$; $n_* = n$;
12:         found = False; // updated minimum
13:     **else**
14:         found = True; // passed minimum
15: **until** found
16: Train the ensemble of $n_*$ networks of size $B/n_*$.
___

accurate prediction of $\text{CNLL}_n$ allows predicting the optimal memory splits for different memory budgets $B$, see the right pairs of plots in figures 6 and 20.

**Finding optimal memory splits in practice.** Let's say we have a budget $B$ and want to find an optimal memory split (MS). Algorithm 1 describes the procedure of finding the optimal MS using the discovered power laws. With this algorithm, we need to train $\min(n, 6)$ networks of size $B/n$ for $n = 1, \ldots, n^* + 1$ where $n^*$ is a number of networks in an optimal MS, and after finding $n^*$, we also need to train lacking networks for the optimal MS. If we do not use power-law predictions we can train MSs one by one (one network of size $B$, then two networks of size $B/2$, etc.) while the quality of the MS starts to degrade. In this case we need to train $n$ networks of size $B/n$ for $n = 1, \ldots, n^* + 1$. As a result, if $n^* \geqslant 4$, power-law predictions allow training fewer networks, and the higher $n^*$ the higher the gain.

## L   Additional experiments with ImageNet

Ashukha et al. [2] released the weights of the ensembles of ResNet50 networks (commonly used size) trained on ImageNet. We used their data to empirically confirm the power-law behaviour of NLL and CNLL as functions of the ensemble size. Figure 21 shows that the resulting power law approximations fit the data well, supporting the results presented in section 4.

Figure 21: Non-calibrated NLL and CNLL of the ResNet50 of the commonly used size on ImageNet. Both $\text{NLL}_n$ and $\text{CNLL}_n$ can be closely approximated with a power law.

## Footnotes

[1] As $\delta < 1$, $\varepsilon_n \to 0$ monotonically and from some $n$ the condition $0 < \varepsilon_n < \mu$ is fulfilled.

[2]We use the same value of $\tau$ for all member networks.