[Reviews · NeurIPS 2020]

Review 1

Summary and Contributions: The paper does a thorough empirical study on the relationship of the number of networks in an ensemble, and the size of each network in an ensemble, and how they relate to each other. The paper focuses on the calibrated NLL scores of the ensembles and tries to find optimal size and number of models in an ensemble given some fixed network training budget. The paper relates each of the findings to the power law while varying the size of the models and the number of the models. The paper systematically justifies the experiments on two different architectures (VGG, WideResNets) and one dataset (CIFAR100).

Strengths: - The paper is very well motivated. The use of ensembles is very important for many reasons, and studying the power law under a given memory budget is very important to the field. - The paper studies the relation between CNLL and the size of the model and the number of models in the parameters, and relates it to the power law. - Although the paper doesn't propose anything novel, the paper is clearly valuable as an empirical study in the field. - The study does very thorough explanation and evaluation and presents the paper in a clear and interpretable manner. - The paper does a very thorough and systematic analysis of the models over the CNLL metric. - To my knowledge, this is the only paper of such kind.

Weaknesses: - I would like to see a more thorough related work. To do a thorough empirical study, it is very important to also cite the important related works. I would like for the authors to consider citing the following papers, which are important for the study of deep ensembles [1,2,3,4,5,6,7,8] - I would like the authors to discuss in the paper on the relation between the ensemble architecture considered in the paper and in [2] as they used shared encoders and n-heads. - Another important thing to comment on is the relation between the number of models as n->\inf and how the model approaches a Bayesian NN [see 4,5] - Many recent papers have also suggested the importance of diversity in NNs as otherwise it can cause the NNs to collapse and be reduced to the same output space [2,4,5] (and [9] which is a more classical take on the subject). The results seem to suggest that even if that is a problem that exists, the models are still able to learn meaningful uncertainty. Why might that be? - It would also be good if the authors considered adding another dataset along with CIFAR-100, as CIFAR-100 is a dataset of 32*32 image patches. - The authors correctly recognize CNLL as an important metric for ensembles, but it would also be interesting to see what the performance of the models are on some benchmark OOD detection tasks (such as the ones considered in [2]). It is not imperative, however showing that would definitely add more interest to the paper. - It is also important to do a timing analysis on how long it takes to train n-models of some size and how that varies. A benchmarked timing analysis is very important since training more networks and deeper networks obviously comes at a cost, which should be discussed in an empirical study. [1] https://arxiv.org/abs/1804.00382 [2] https://arxiv.org/abs/2003.04514 [3] https://arxiv.org/abs/1912.02757 [4] https://arxiv.org/abs/2001.10995 [5] https://arxiv.org/abs/2002.08791 [6] http://papers.nips.cc/paper/7219-simple-and-scalable-predictive-uncertainty-estimation-using-deep-ensembles [7] https://arxiv.org/abs/2005.07292 [8] http://papers.nips.cc/paper/6270-stochastic-multiple-choice-learning-for-training-diverse-deep-ensembles [9] https://dl.acm.org/doi/abs/10.1145/1015330.1015385?casa_token=HMjBA3QbKlkAAAAA:OByAELh4KVDkhVXcCYUyQzO9sy8yaQtOggxYv1cv6oqBm84Vhb_mV8VBOdGvYoWHEsLB1LtojV7Aog

Correctness: - The paper is an empirical analysis, and it does a proper job supporting the experiments and the claims are well-founded. - I agree with all the claims and methodology used in the paper. The paper is thorough in validating claims and chooses an appropriate metric to perform the empirical analysis.

Clarity: The paper is very well written and the experiments are well presented in the plots. But the paper needs significant improvements in the related works section. Empirical studies should have a strong related works section as they serve to represent a whole body of work, and the literature on ensembling NNs is very popular (see weaknesses for some suggestions).

Relation to Prior Work: The paper is an empirical study, and makes justified choices. The paper mentions concurrent work but the work seems to be sufficiently different.

Reproducibility: Yes

Additional Feedback: Happy to increase my score further, if the authors incorporate my suggestions in the rebuttal. POST REBUTTAL: I have decided to stick with my score after reading the rebuttal + other reviews. The rebuttal sufficiently satisfies my points, although I don't agree with the points regarding bagging reducing the uncertainty estimation, since I do not believe that to be the case. As the functional diversity of the ensembles increase, the uncertainty should not decrease, since ensembles suffer from a notion of "posterior collapse". I suggest the authors to not include that discussion / rephrase that suggestion in the final camera-ready paper. But regardless of that, I do think this paper is very important for the community, and without a doubt should be accepted.


Review 2

Summary and Contributions: This paper looks at the calibrated log likelihood as a proxy for uncertainty estimation of a deep ensemble. It shows that one can fit a power law to the CNLL as a function of ensemble size, network size (width), and total parameter count of the ensemble. This has a number of benefits, including the ability to reason about the uncertainty properties of an infinite ensemble size and to determine the optimal split of ensemble size and network width given a fixed memory budget.

Strengths: This paper is extremely thorough and addresses an important practical problem with neural networks, which is that they provide poor uncertainty estimates. This paper is I believe the first to provide tools for predicting the uncertainty properties of an ensemble, and to even make theoretical claims about the performance of an infinitely large ensemble. I'd hope that this provides building blocks for future work along the lines of what Neural Tangent Kernel was able to do with infinite width neural networks.

Weaknesses: The entire work is somewhat premised on the idea that CNLL is a good proxy for uncertainty. While this might be a valid assumption, it would be nice to see the results empirically validated by looking at other potential metrics, such as ECE or TACE to measure calibration, and maybe even look at distribution shift using a dataset like CIFAR10-C.

Correctness: As far as I can tell, the experiments are very thorough and methodologically correct. The original DeepEnsembles paper did have an adversarial training component to it, which doesn't appear to be part of this implementation. I don't think that's a substantial problem, but it might be good to mention the difference.

Clarity: Generally it is very well written. There are a couple typos/grammatical errors sprinkled throughout, so it would be good to take a proofreading pass.

Relation to Prior Work: Yes, as far as I'm aware.

Reproducibility: Yes

Additional Feedback: AFTER REBUTTAL: After seeing other reviewers takes and the authors' response, I am dropping slightly to an 8. I still think this is a very strong and valuable paper, but maybe not as earth-shattering as my initial review indicated.


Review 3

Summary and Contributions: The authors study the dependence of the calibrated negative log likelihood of an ensemble of networks as a function of the ensemble size n and a the network size s (width in particular). The propose to model them as powerlaws, give a theoretical argument for why that is appropriate, and observe the behavior of real models (VGG and WideResNet) on CIFAR-100. Based on the fitted powerlaws, they infer the optimal split between model size s and number of models n, make a prediction, and verify its agreement with empirical observations.

Strengths: Overall I find the paper well written, interesting, and attacking a really important question. I have been wondering about a similar question for some time and I am happy to see a very good attempt at resolving it. The question is clear, well-specified, and well-answered. I particularly appreciate that the authors test their memory split prediction against empirical observations. I think this is a good paper overall.

Weaknesses: 1) How valid / necessary is the power-law fit? Looking at the values in Figure 1, the range of negative log likelihoods in the leftmost panel is 0.8 -1.8. The middle panel show the linear scale and beyond the n = 1 point, the curves look pretty straight to me. That is reflected in the right panel where you show the index of the powerlaw is just a bit above -1.0, so essentially 1/n and a bit, but a bit slower decay. Many functions would locally look like this. What are the main reasons for using the powerlaw fit in particular? 2) Derivation in equation 2. What would be the conditions on the derivation in equation 2 to work? You characterize the distribution by its first two moments, which seems fine, but how sensitive is your conclusions that the resulting NLL depends on 1/n (=> the powerlaw) on the your assumption about the distribution of p*? In particular since you work with the log of those, will there be any other additional complications caused by that? 3) Different architectures, different power-laws? I wonder how to use this in practice given that different architectures seem to produce different powerlaws. Would I have to do an exploration of the (n,s) plane first, make a fit, and only then be able to predict the optimal memory split? This could defeat the practical purpose if that exploration took a lot of resources.

Correctness: The paper seems correct and sound as much as I can say.

Clarity: The paper is clearly written.

Relation to Prior Work: The relevant work seems broadly mentioned, however, I am not an expert in the negative log likelihood of ensembles in particular, so I cannot judge the completeness.

Reproducibility: Yes

Additional Feedback: POST REBUTTAL: The authors addressed my questions well and I will keep my score at 7 = accept. I really like the paper -- it's message is clear and I think it'll be useful to the field.


Review 4

Summary and Contributions: Investigate presence of power laws in calibrated negative log likelihood (CNNL) of deep ensembles. Theoretically motivate their presence and verify experimentally. Show that the power laws can be used to predict CNNL of different memory splits (several medium-size networks while keeping the total number of parameters constant) and thus finding the optimal split.

Strengths: - Significance: Prediction based on the observed power laws is useful in choosing deep ensemble memory splits - Theoretical grounding and empirical evaluation look good, but haven't checked in detail - Provides more data on the double descent behavior, which authors find not to occur in CNNL compared to NNL (Figure 4).

Weaknesses: No explanation for why calibration removes the double-descent behaviour

Correctness: Yes

Clarity: Yes

Relation to Prior Work: Yes

Reproducibility: Yes

Additional Feedback: Please include the code in the supplementary material

[Author Response · NeurIPS 2020]

We thank all the reviewers for such thoughtful and high-quality reviews as well as for the positive feedback! We will do
our best to incorporate all the recommendations in the next revision of the paper.

**R1, R3: Experiments on more datasets.** We would like to note that we performed all experiments not only on
CIFAR100 but also on CIFAR10 dataset, as mentioned in the experimental setup (line 129). All results are similar to
the ones on CIFAR100 and presented in appendices. We also experimented with ResNet50 models on ImageNet, but
only of standard size due to the computational limitations, therefore we did not include them in the paper. We analysed
the behaviour of NLL and CNLL of the models as functions of ensemble size (as in section 4) and the results are the
same as the ones in the paper. We will add the results in the appendix.

**R1: More thorough related work.** We kindly thank the reviewer for the provided references. We had to reduce the
related work section due to space limitation but will revise it and make it more broad and thorough in the next version
of the paper. Below we comment on the specific related topics mentioned by R1. References are numbered according to
R1.

Relation to different ensemble technique [2]. In our work, we consider simple ensembles of neural networks indepen-
dently trained from random initializations, as they are widely used in practice. Different ensembling modifications, e.g.
proposed in [1, 2], go beyond the scope of our work, however, we regard this direction as very interesting.

Relation to Bayesian NNs. Despite the asymptotic nature of our theoretical results, we mainly consider finite ensembles
in our experiments. The Bayesian (and other limiting) perspective on ensembling and its connection to our results for
$n \to \infty$ is a promising direction for future research.

On model diversity in ensembles. This issue was partially addressed in [6, 3], where it is shown that random initialization
allows to explore different modes in function space, which explains why deep ensembles trained with just random
initialization perform well in practice. Moreover, as noted in [6], simple diversity inducing techniques, like bagging,
may even deteriorate performance in uncertainty estimation.

**R1, R2: Other metrics/benchmarks for uncertainty estimation.** Considering other metrics/benchmarks is a
direction for future research since it requires a separate thorough study, as different metrics have various specifics.

**R1: Timing analysis.** For a single VGG on CIFAR-100, for network sizes 0.125 / 0.25 / 0.5 / 1 / 2 / 4 / 8 (in standard
budgets), testing takes 6.8 / 9.5 / 20 / 33 / 63 / 111 / 227 seconds (batch size 1024), while one training epoch takes 7.8 /
8 / 11 / 16 / 27 / 42 / 80 seconds (batch size 64). For example, for budget $4S$, with a single network / memory split of 4
networks, testing takes 111 / 132 sec, while one training epoch takes 42 / 64 seconds. So training and using memory
split is slower than a single network, but only moderately, not four times slower.

**R3: Why do we use power law?** From the start, we had two main motivations: 1) the theoretical one — theoretical
results presented in the paper show the asymptotic power-law behavior of (C)NLL, 2) the practical one — plots of
(C)NLL do look similar to power laws in practice. In our experiments, power laws allow both accurate *interpolation*
and *extrapolation*. Interpolation: we can approximate (C)NLL with a power law *on a wide range of arguments*.
Extrapolation: (C)NLL fitted on a *short* segment approximates well the *full* range of arguments. Hence, we argue that
(C)NLL follows a power law and not another function.

**R3: The conditions for derivation in equation 2.** The only condition on the distribution of ensemble predictions
$p^*$ required for the derivation of equation 2 (besides i.i.d.) is separability from zero, i.e. $p^* \in [\epsilon, 1]$, just due to
irregular behavior of logarithm near zero. Given that, we can obtain our main theoretical result stated in Proposition 1 —
asymptotic power-law behavior of the ensemble NLL. The rigorous derivation is provided in Appendix A.1.

**R3: How to use predictions with power laws in practice?** Let's say we have a budget $B$ and want to find an optimal
memory split (MS). We consider MSs with number of networks $n = 1, 2, 4, 8, \dots$. If we do not use power-law
predictions we can train MSs one by one (one network of size $B$, then two networks of size $B/2$ and so on) while the
quality of the MS starts to decrease. In this case we need to train $n$ networks of size $B/n$ for $n = 1, \dots, n^* + 1$, where
$n^*$ is a number of networks in an optimal MS. If we use power-law predictions in the same manner we did in the paper
(see section 7) we need to train $\min(n, 6)$ networks of size $B/n$ for $n = 1, \dots, n^* + 1$ and then use predictions. After
finding $n^*$ we also need to train lacking networks for the optimal MS. As a result, if $n^* \geq 4$, power-law predictions
allow training fewer networks, and the higher $n^*$ the higher the gain.

**R4: Why calibration removes the double-descent behavior?** The accurate answer to this question requires a more
thorough study of double descent. Our preliminary experiments show that model overfits in terms of NLL easier than in
terms of accuracy or CNLL, therefore double descent of NLL can be observed more often.

[Meta-Review · NeurIPS 2020]

The paper investigates the performance of deep ensembles as a function of number of ensemble members, and identifies conditions under which calibrated negative log-likelihood follows a power law behavior. Given the increasing popularity of deep ensembles, this a timely work and all the reviewers recommend acceptance. The reviewers raised questions about missing references, experiments on additional datasets and how to use this in practice. The author rebuttal and the proposed revisions to the camera ready mostly address these concerns. Overall, this is a good paper and I recommend acceptance.